# Visual analysis of mass cytometry data by hierarchical stochastic neighbour embedding reveals rare cell types

Vincent van Unen [1], Thomas Höllt[2,3], Nicola Pezzotti[2], Na Li[1], Marcel J.T. Reinders [4], Elmar Eisemann[2], Frits Koning[1], Anna Vilanova[2] & Boudewijn P.F. Lelieveldt[4,5]

Mass cytometry allows high-resolution dissection of the cellular composition of the immune system. However, the high-dimensionality, large size, and non-linear structure of the data poses considerable challenges for the data analysis. In particular, dimensionality reduction-based techniques like t-SNE offer single-cell resolution but are limited in the number of cells that can be analyzed. Here we introduce Hierarchical Stochastic Neighbor Embedding (HSNE) for the analysis of mass cytometry data sets. HSNE constructs a hierarchy of non-linear similarities that can be interactively explored with a stepwise increase in detail up to the single-cell level. We apply HSNE to a study on gastrointestinal disorders and three other available mass cytometry data sets. We find that HSNE efficiently replicates previous observations and identifies rare cell populations that were previously missed due to downsampling. Thus, HSNE removes the scalability limit of conventional t-SNE analysis, a feature that makes it highly suitable for the analysis of massive high-dimensional data sets.

[1] Department of Immunohematology and Blood Transfusion, Leiden University Medical Center, Albinusdreef 2, 2333 ZA Leiden, The Netherlands. [2] Computer Graphics and Visualization Group, Delft University of Technology, Mekelweg 4, 2628 CD Delft, The Netherlands. [3] Computational Biology Center, Leiden University Medical Center, Albinusdreef 2, 2333 ZA Leiden, The Netherlands. [4] Pattern Recognition and Bioinformatics Group, Delft University of Technology, Mekelweg 4, 2628 CD Delft, The Netherlands. [5] Division of Image Processing, Department of Radiology, Leiden University Medical Center, Albinusdreef 2, 2333 ZA Leiden, The Netherlands. Vincent van Unen, Thomas Höllt and Nicola Pezzotti contributed equally to this work. Frits Koning, Anna Vilanova and Boudewijn P.F. Lelieveldt jointly supervised this work. Correspondence and requests for materials should be addressed to V.v.U. (email: V.van_unen@lumc.nl) or to B.P.F.L. (email: B.P.F.Lelieveldt@lumc.nl)

Mass cytometry (cytometry by time-of-flight; CyTOF) allows the simultaneous analysis of multiple cellular markers (>30) present on biological samples consisting of millions of cells. Computational tools for the analysis of such data sets can be divided into clustering-based and dimensionality reduction-based techniques[1], each having distinctive advantages and disadvantages. The clustering-based techniques, including SPADE[2], FlowMaps[3], Phenograph[4], VorteX[5] and Scaffold maps[6], allow the analysis of data sets consisting of millions of cells but only provide aggregate information on generated cell clusters at the expense of local data structure (i.e., single-cell resolution). Dimensionality reduction-based techniques, such as PCA[7], t-SNE[8] (implemented in viSNE[9]), and Diffusion maps[10], do allow analysis at the single-cell level. However, the linear nature of PCA renders it unsuitable to dissect the non-linear relationships in the mass cytometry data, while the non-linear methods (t-SNE[8] and Diffusion maps[10]) do retain local data structure, but are limited by the number of cells that can be analyzed. This limit is imposed by a computational burden but, more importantly, by local neighborhoods becoming too crowded in the high-dimensional space, resulting in overplotting and presenting misleading information in the visualization. In cytometry studies, this poses a problem, as a significant number of cells needs to be removed by random downsampling to make dimensionality reduction computationally feasible and reliable. Future increases in acquisition rate and dimensionality in mass- and flow cytometry are expected to amplify this problem significantly[11,12].

Here we adapted Hierarchical stochastic neighbor embedding (HSNE)[13] that was recently introduced for the analysis of hyperspectral satellite imaging data to the analysis of mass

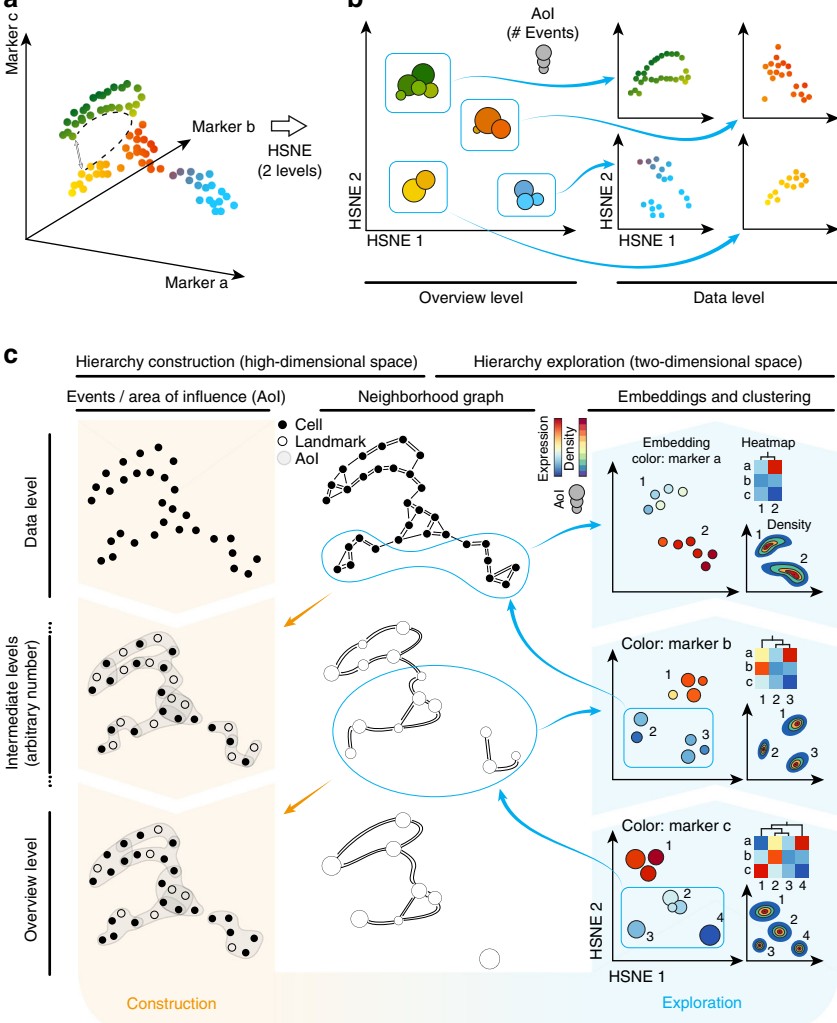

**Fig. 1** Schematic overview of Cytosplore+HSNE for exploring the mass cytometry data. By creating a multi-level hierarchy of an illustrative 3D data set (**a**), we achieve a clear separation of different cell groups in an overview embedding (left panel **b**) that conserves non-linear relationships (i.e., follows the distance indicated by the dashed line in **a**, instead of the grey arrow) and more detail within the separate groups on the data level (right panel **b**). **c** Construction and exploration of the hierarchy. The hierarchy is constructed starting with the data level (left two columns). On the basis of the high-dimensional expression patterns of the cells, a weighted kNN graph is constructed, which is used to find representative cells used as landmarks in the next coarser level. By administering the area of influence (AoI) of the landmarks, cells/landmarks can be aggregated without losing the global structure of the underlying data or creating shortcuts. The exploration of the hierarchy is shown in the two rightmost columns. At the bottom, we see the overview level (in this example the 3rd level in the hierarchy), which shows that a group of landmarks has low expression in marker c (bottom-right panel). Selecting this group of landmarks for further exploration results in a look-up of the landmarks in the preceding level (neighborhood graph, intermediate level) that are in the AoI, with which a new embedding can be created at the 2nd level of the hierarchy (middle-right panel). Marker b shows a strong separation between the upper and lower landmarks at this level. Zooming-in on the landmarks with low expression of marker b reveals further separation in marker a at the lowest level, the full data level (top-right panel)

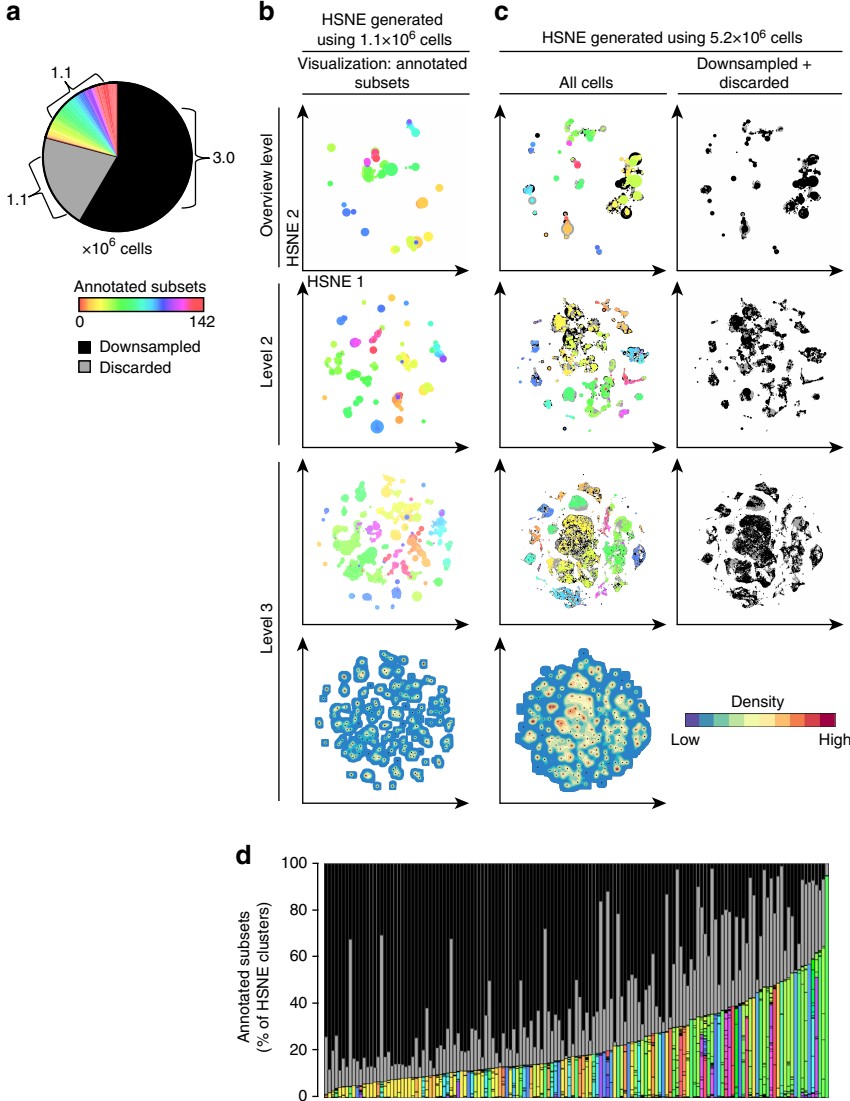

**Fig. 2** Gain of information by analyzing the mass cytometry data at full resolution with Cytosplore[+HSNE]. **a** Pie chart showing cellular composition of the mass cytometry data set. Color represents the subsets ($N = 142$), as identified in our previous study[14]. Black represents the cells discarded by stochastic downsampling and grey represents the cells discarded by ACCENSE clustering. **b** Embeddings of the 1.1 million cells annotated in ref [14] showing the top three levels of the HSNE-hierarchy (five levels in total). Color represents annotations as in **a**. Size of the landmarks is proportional to the number of cells in the AoI that each landmark represents. Bottom map shows density features depicting the local probability density of cells for the level 3 embedding, where black dots indicate the centroids of identified cluster partitions using GMS clustering. **c** Embeddings of all 5.2 million cells, again showing only the top three levels of the hierarchy (five levels in total). Colors as in **a**. Right panels visualize landmarks representing cells discarded by stochastic downsampling (black) and the cells discarded by ACCENSE (grey). Bottom map shows density features for the level 3 embedding as described in (**b**). **d** Frequency of annotated cells for 145 clusters identified by Cytosplore[+HSNE] at the third hierarchical level using GMS clustering in **c**. Color coding as in **a**

cytometry data sets to visually explore millions of cells while avoiding downsampling. HSNE builds a hierarchical representation of the complete data that preserves the non-linear high-dimensional relationships between cells. We implemented HSNE in an integrated single-cell analysis framework called Cytosplore [+HSNE]. This framework allows interactive exploration of the hierarchy by a set of embeddings, two-dimensional scatter plots where cells are positioned based on the similarity of all marker expressions simultaneously, and used for subsequent analysis such as clustering of cells at different levels of the hierarchy. We found that Cytosplore[+HSNE] replicates the previously identified hierarchy in the immune-system-wide single-cell data[4,5,14], i.e., we can immediately identify major lineages at the highest overview level, while acquiring more information by dissecting the

immune system at the deeper levels of the hierarchy on demand. Additionally, Cytosplore[+HSNE] does so in a fraction of the time required by other analysis tools. Furthermore, we identified rare cell populations specifically associating to diseases in both the innate and adaptive immune compartments that were previously missed due to downsampling. We highlight scalability and generalizability of Cytosplore[+HSNE] using three other data sets, consisting of up to 15 million cells. Thus, Cytosplore[+HSNE] combines the scalability of clustering-based methods with the local single-cell detail preservation of non-linear dimensionality reduction-based methods. Finally, Cytosplore[+HSNE] is not only applicable to mass cytometry data sets, but can be used for the other high-dimensional data like single-cell transcriptomic data sets.

## Results

**Hierarchical exploration of massive single-cell data**. For a given high-dimensional data set such as the three-dimensional illustrative example in Fig. 1a, HSNE[13] builds a hierarchy of local neighborhoods in this high-dimensional space, starting with the raw data that, subsequently, is aggregated at more abstract

hierarchical levels. The hierarchy is then explored in reverse order, by embedding the neighborhoods using the similarity-based embedding technique, Barnes–Hut (BH)-SNE[15]. To allow for more detail and faster computation, each level can be partitioned in part or completely, by manual gating or unsupervised clustering, and partitions are embedded separately on the next, more

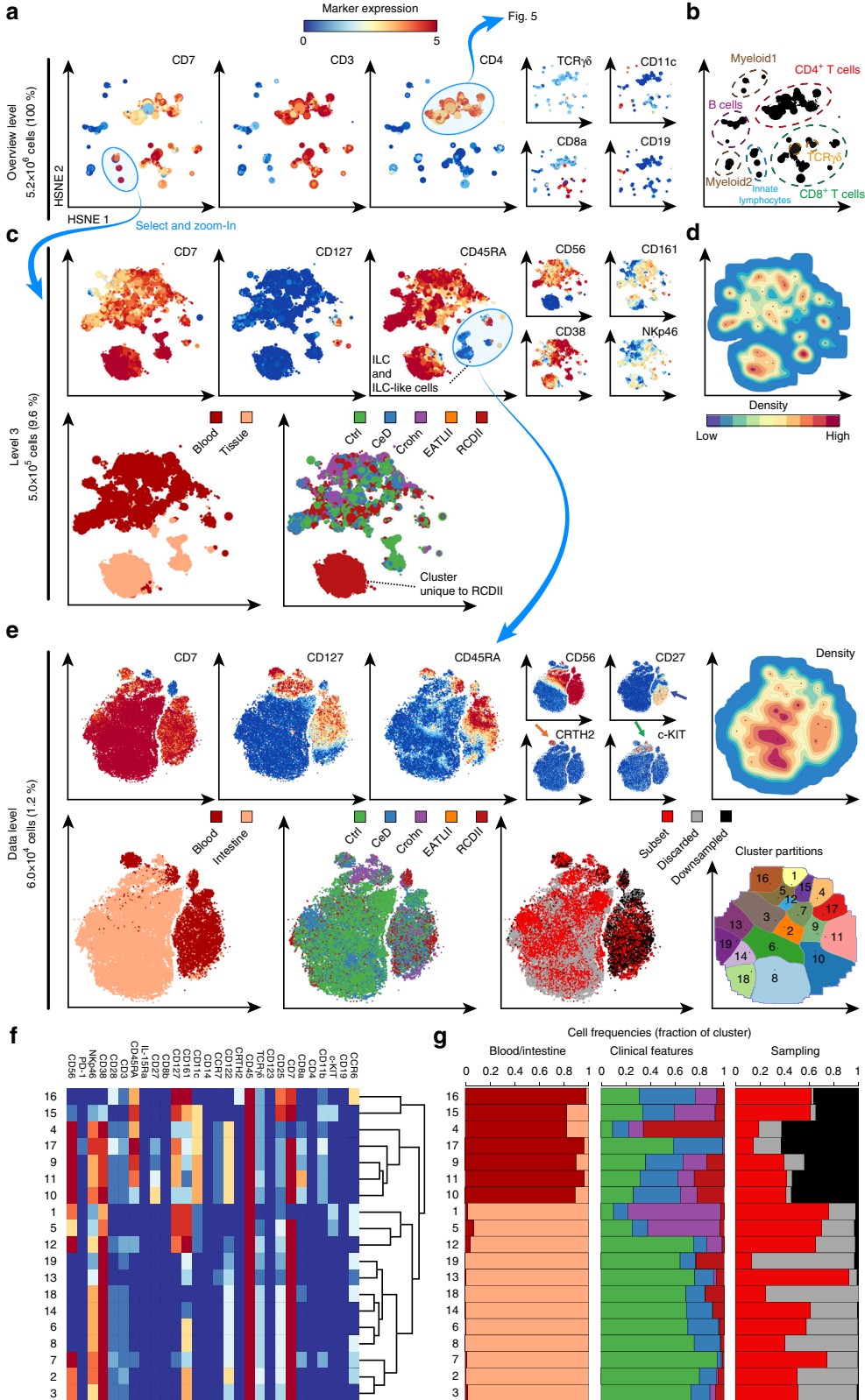

detailed level (compare Fig. 1b). HSNE works particularly well for the analysis of the mass cytometry data because the local neighborhood information of the data level is propagated through the complete hierarchy. Groups of cells that are close in the Euclidian sense (Fig. 1a, grey arrow), but not on the non-linear manifold (Fig. 1a, dashed black line), are well separated even at higher aggregation levels (Fig. 1b). The power of HSNE lies in its scalability to tens of millions of cells, while the possibility to continuously explore the hierarchy allows the identification of rare cell populations at the more detailed levels. Next follows a general description of how the hierarchy is built and explored through embeddings. More details can be found in the Methods section.

The left panels of Fig. 1c give an overview of the HSNE-hierarchy construction. We show the hierarchy from the fine-grained data level to an overview level from the top to bottom panels. The number of levels is defined by the user and depends mostly on the input-data size. While the data aggregation is completely data-driven, for a typical mass cytometry data set, every additional level reduces the number of landmarks by roughly one order of magnitude. Therefore, we recommend to use log10(N/100) levels, with N being the number of cells: this generally results in at most few thousands of landmarks at the highest level of the hierarchy. The foundation of the hierarchy is constructed using the original input data. Each dot represents a single cell (Fig. 1c, data level). Similarities between cells on the data level are defined by building an approximated, weighted k-nearest neighbor (kNN) graph[16] using the Euclidian distances based on the complete marker expression (Fig. 1c, top-center panel). The weights of this graph can directly be used as input to embed the data into a two-dimensional space (Fig. 1c, top-right panel). With the BH-SNE the two-dimensional embedding is generated such that the layout of the points indicates similarities between the cells in the high-dimensional space according to the neighborhood graph.

To aggregate the data into the next level (Fig. 1c, intermediate levels), we identify representative cells to use as landmarks (Fig. 1c, white circles). For that, the weighted kNN graph is interpreted as a Finite Markov Chain and the most influential (i.e., best-connected) nodes are chosen as landmarks, using a Monte Carlo process. The landmarks are then embedded into a two-dimensional space based on their similarities. However, simply repeating the kNN construction with Euclidian distances for the selected landmarks in the high-dimensional space would eventually eliminate non-linear structures by creating undesired "shortcuts" in the graph (a problem reported by Setty et al.[17] in a different setting). Instead, we define the area of influence (AoI) of each landmark, indicated by the grey hulls (Fig. 1c, left panels), as the cells that are well-represented by the landmark according to the kNN graph. Different landmarks can have overlapping regions of locally-similar cells. Therefore, we define the similarity of two landmarks as the overlap of their respective AoIs. Furthermore, we construct a neighborhood graph, based on these similarities. Here, two nodes are connected if they have overlapping AoIs. The strength of the connection is defined by the number of data points within the overlapping region. This graph replaces the kNN graph as input for levels subsequent to the data level. Hereby, we effectively maintain the non-linear structure of the data to the top of the hierarchy and avoid shortcuts (Fig. 1c, bottom panels). We show that the preservation of non-linear neighborhoods by HSNE indeed conserves structure that is otherwise lost by random downsampling (Supplementary Note 1. Cytosplore$^{+HSNE}$ is reproducible and robust. and Supplementary Fig. 1).

The data exploration in Cytosplore$^{+HSNE}$ starts with the visualization of the embedding at the highest level, the overview level (Fig. 1c, bottom-right panel). Similar to other embedding techniques for visualizing the single-cell data[4,9], the layout of the landmarks indicates similarity in the high-dimensional space according to the level's neighborhood graph. Color is used to represent additional traits, such as marker expressions. The landmark size reflects its AoI. While it is possible to continuously select all landmarks and compute a complete embedding of the next, more detailed level, this strategy would eventually embed all the data and suffer from the same scalability problems as a t-SNE embedding, i.e., overcrowding (Supplementary Note 2. Millions of cells cause performance issues and overcrowding in t-SNE. and Supplementary Fig. 2) and slow performance. Instead, we envision that the user selects a group of landmarks, by manual gating based on visual cues such as patterns found in marker expression, or by performing unsupervised Gaussian mean shift (GMS) clustering[18] of the landmarks based on the density representation of the embedding (Fig. 1c, right panels). Then, the user can zoom into this selection by means of a more detailed embedding. This means that, all landmarks/cells in the combined AoI on the preceding level are retrieved from the neighborhood graph (Fig. 1c, blue encirclements), embedded, and visualized in a new view. Moreover, interactively linked heatmap visualizations of clusters (Fig. 1c, right panels) and descriptive statistics of markers within a selection can be used to guide the exploration. For example, these tools allow to inspect the heterogeneity of cells within individual clusters, including the cells associated to individual landmarks. Importantly, all of the described tools are available at every level of the hierarchy and linked interactively. Selections in the embedding and heatmap at one level of the hierarchy can thus be highlighted in the embeddings of other levels (Supplementary Fig. 3). All these aspects are further demonstrated using a typical exploration workflow with Cytosplore$^{+HSNE}$ in the Supplementary Movie 1. With this strategy, tens of millions of cells can be explored, providing both global visualizations up to single-cell resolution visualizations, while preserving non-linear relationships between landmarks/cells at all levels of the hierarchy.

**HSNE eliminates the need for downsampling.** In a previous study[14], a mass cytometry data set on 5.2 million cells derived from intestinal biopsies and paired blood samples was analyzed using a SPADE-t-SNE-ACCENSE pipeline. Due to t-SNE

**Fig. 3** Analysis of the CD7$^+$CD3$^-$ innate lymphocyte compartment in inflammatory intestinal diseases. **a** First HSNE level embedding of 5.2 million cells. Color represents arcsin5-transformed marker expression as indicated. Size of the landmarks represents AoI. Blue encirclement indicates selection of landmarks representing CD7$^+$CD3$^-$ innate lymphocytes and CD4$^+$ T cells further discussed in Fig. 5. **b** The major immune lineages, annotated on the basis of lineage marker expression. **c** Third HSNE level embedding of the CD7$^+$CD3$^-$ innate lymphocytes ($5.0 \times 10^5$ cells). Color represents arcsin5-transformed marker expression in top panels, and tissue-origin and clinical features in bottom panels. Blue encirclement indicates selection of landmarks representing CD127$^+$ILC and ILC-like cells. **d** Third HSNE level embedding shows density features depicting the local probability density of cells, where black dots indicate the centroids of identified cluster partitions using GMS clustering. **e** Embedding of the CD127$^+$ILC and ILC-like cells ($6.0 \times 10^4$ cells) at single-cell resolution. Arrows indicate ILC1 (blue), ILC2 (orange) and ILC3 (green). Bottom-right panel shows corresponding cluster partitions using GMS clustering based on density features (top-right panel). **f** A heatmap summary of median expression values (same color coding as for the embeddings) of cell markers expressed by CD127 + ILC and ILC-like clusters identified in **b** and hierarchical clustering thereof. **g** Composition of cells for each cluster is represented graphically by a horizontal bar in which segment lengths represent the proportion of cells with: (left) tissue-of-origin, (middle) disease status and (right) sampling status

| Subset | Phenotype | Annotation |
|---|---|---|
| 16 | CD127$^+$CD161$^+$CD25$^+$CD122$^-$CRTH2$^+$ | ILC2 |
| 15 | CD127$^+$CD161$^+$CD25$^+$CD122$^-$CRTH2$^-$ | ILC2-like |
| 4 | CD56$^+$NKp46$^+$CD127$^-$CD161$^-$c-KIT$^-$ | NK-like |
| 17 | CD56$^+$NKp46$^+$CD127$^+$CD161$^-$c-KIT$^-$ | ILC1-like |
| 9 | CD56$^+$NKp46$^+$CD127$^+$CD161$^-$c-KIT$^-$ | ILC1-like |
| 11 | CD56$^+$NKp46$^+$CD127$^+$CD161$^-$c-KIT$^-$ | ILC1-like |
| 10 | CD56$^+$NKp46$^+$CD127$^-$CD161$^-$c-KIT$^-$ | NK-like |
| 1 | CD7$^-$CD127$^+$CD161$^+$c-KIT$^+$ | ILC3-like |
| 5 | CD7$^+$CD127$^+$CD161$^+$c-KIT$^+$ | ILC3 |
| 12 | CD56$^+$CD127$^+$CD161$^+$c-KIT$^-$CD27$^-$ | ILC1-like |
| 19 | CD56$^-$CD127$^-$NKp46$^-$CD161$^{dim}$ | Lin- cells |
| 13 | CD56$^-$CD127$^-$NKp46$^-$CD161$^{dim}$ | Lin- cells |
| 18 | CD56$^-$CD127$^-$NKp46$^+$CD161$^-$ | Lin- cells |
| 14 | CD56$^-$CD127$^-$NKp46$^+$CD161$^-$ | Lin- cells |
| 6 | CD56$^-$CD127$^-$NKp46$^+$CD161$^+$ | Lin- cells |
| 8 | CD56$^-$CD127$^-$NKp46$^+$CD161$^+$ | Lin- cells |
| 7 | CD56$^+$CD127$^-$CD45RA$^-$CD161$^-$ | NK-like |
| 2 | CD56$^+$CD127$^-$CD45RA$^-$CD161$^+$ | NK-like |
| 3 | CD56$^+$CD127$^-$CD45RA$^-$CD161$^+$ | NK-like |

**Fig. 4** CD127$^+$ILC and ILC-like subsets identified by Cytosplore$^{+HSNE}$. Table showing cluster number, distinguishing phenotypic marker expression profiles and biological annotation for the clusters identified in Fig. 3e. Black color indicates clusters described in previous reports and red color additional unknown clusters. Hierarchical clustering of clusters based on marker expression profile shown in the heatmap depicted in Fig. 3f

limitations, the data set had to be downsampled by 57.7% (Fig. 2a), where it was decided to equal the number of cells from blood and intestinal samples for a balanced comparison, which led to the exclusion of more cells from the blood samples. Moreover, ACCENSE clustered only 50% of the t-SNE-embedded data into subsets (Fig. 2a). Together, this excluded 78.8% of the cells from the analysis. The remaining 1.1 million cells were annotated into 142 phenotypically distinct immune subsets[14] (Fig. 2a).

To determine whether Cytosplore$^{+HSNE}$ could identify similar subsets, we embedded the 1.1 million annotated cells (Fig. 2b). Computation time was in the order of minutes and the analysis was finished within an hour, compared to 8 weeks of computation in the original study. Color coding shows the grouping of subsets at all hierarchical levels. GMS clustering at the third level embedding (Fig. 2b, bottom panel) reveals that 75.5% of cells were assigned to a single subset by both methods (Supplementary Fig. 4). Hence, to reach similar results it was not necessary to explore the data at lower (more detailed) levels.

Next, we utilized Cytosplore$^{+HSNE}$ to analyze the complete dataset on 5.2 million cells, thus including the cells that were discarded in the SPADE-t-SNE-ACCENSE pipeline. The embeddings show by color coding that subsets of the same immune lineage clustered at all three levels (Fig. 2c). More interestingly, the cells removed during downsampling (shown in black) and cells ignored during the ACCENSE clustering (shown in grey) were positioned throughout the entire map (Fig. 2c). We selected 145 clusters using GMS clustering at the third level and observed that the identified clusters contained variable numbers of downsampled and non-classified cells (Fig. 2d). These findings indicate that both the non-uniform downsampling and the cell losses during the ACCENSE clustering introduce a potential bias in observed heterogeneity in the immune system. Cytosplore$^{+HSNE}$ overcomes this problem as it analyzes all cells and does so efficiently.

**HSNE identifies rare subsets in the ILC compartment**. We illustrate an exploration workflow with Cytosplore$^{+HSNE}$ using the data set of 5.2 million cells[14] (Fig. 3). At the overview level, 4090 landmarks depict the general composition of the immune system (Fig. 3a) and color coding is applied to reveal CD-marker expression patterns on the basis of which the major immune lineages are identified (Fig. 3b). Next the CD7$^+$CD3$^-$ cell clusters were selected as indicated and a new higher resolution embedding was generated at level 3 of the hierarchy (Fig. 3c). Here, coloring of the landmarks based on marker expression (Fig. 3c, top panels) and a density plot of the embedding is shown (Fig. 3d) alongside the clinical features of the subjects from which the samples were obtained and the tissue-origin of the landmarks (Fig. 3c, bottom panels). This reveals a cluster of cells abundantly present in the intestine of patients with refractory celiac disease (RCDII). In addition, a large cluster of CD45RA$^+$CD56$^+$ NK cells and three distinct innate lymphoid cell (ILC) clusters with a characteristic lineage$^-$ CD7$^+$CD161$^+$CD127$^+$ marker expression profile[19,20] are visualized. Strikingly, a distinct population of CD7$^+$CD127$^-$CD45RA$^-$ and partly CD56$^+$ cells is found in between the NK, RCDII and ILC cell clusters.

To uncover the phenotypes of these ILC-related clusters, we next embedded the ILC and ILC-like clusters (Fig. 3c, selection) at the full single-cell data level (59,775 cells; 1.2% of total) (Fig. 3e). The marker expression overlays revealed that the majority of cells are CD7$^+$ and displayed variable expression levels for CD127, CD45RA, and CD56 (Fig. 3e). In addition, and in line with previous reports[21,22], (co-)expression of CD127 with CD27, CRTH2, and c-KIT revealed the phenotypes corresponding to helper-like ILC type 1, 2 and 3, respectively (indicated by arrows in Fig. 3e). Moreover, by visualizing the tissue-origin in the Cytosplore$^{+HSNE}$ embedding the tissue-specific location of ILC and ILC-related phenotypes became evident (Fig. 3e).

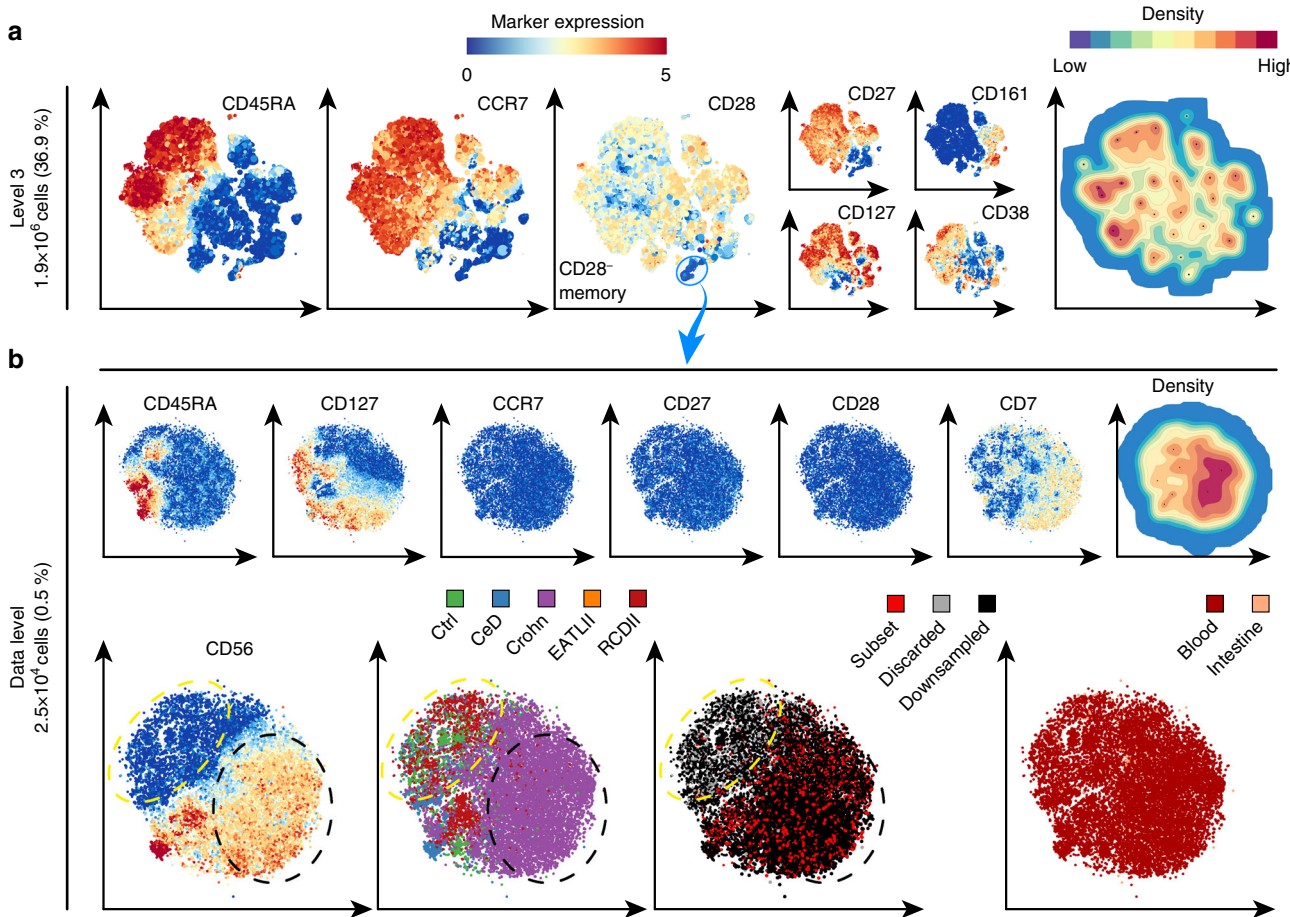

**Fig. 5** Analysis of the CD4[+] T-cell compartment in inflammatory intestinal diseases. **a** Third HSNE level embedding of the CD4[+] T cells ($1.4 \times 10^6$ cells, selected in Fig. 3). Color and size of landmarks as described in Fig. 3. Right panel shows density features for the level 3 embedding. Blue encirclement indicates selection of landmarks representing CD28[−]CD4[+] T cells. **b** Embedding of the CD28[−]CD4[+] T cells ($2.6 \times 10^4$ cells) at single-cell resolution. Bottom-left panel shows yellow and black dashed encirclements based on CD56[−] and CD56[+] expression, respectively. Three bottom-right panels show cells colored according to: (left) from subjects with different disease status (CeD, Crohn, EATLII, RCDII, and controls), (middle) sampling status (annotated subset, discarded by ACCENSE and downsampled) and (right) tissue-of-origin (blood and intestine)

Next, we performed GMS clustering on the full data level embedding, which resulted in 19 phenotypically distinct clusters (Fig. 3e, right plots) based on marker expression profiles (Fig. 3f). The cell surface phenotypes of 8 out of the 19 clusters (Fig. 3f) matched previously described[21] biological annotations (Fig. 4, black annotations) including the CRTH2[+]ILC2 (cluster 16), c-KIT[+]ILC3 (cluster 5) and CD56[−]CD127[−] lineage[−] IELs (cluster 19, 13, 18, 14, 6, and 8), the latter representing innate type of lymphocytes with dual T-cell precursor and NK/ILC traits[23–25]. Remarkably, the remaining 11 clusters strongly resembled distinct ILC types, but did not fulfil the complete phenotypic requirements according to established nomenclature[21] (Fig. 4, red annotations). For example, cluster 15 is highly similar to ILC2 (cluster 16) based on the expression of CD7, CD127, CD161, and CD25, but lacks the ILC2-defining marker CRTH2. Also, clusters 17, 9 and 11 bear close resemblance to ILC1 based on CD7[+]CD127[+]c-KIT[−] marker expression profile, but lack the ILC-defining CD161 marker. Finally, cluster 1 is very similar to ILC3 (cluster 5) based on CD127, CD161 and c-KIT positivity, but lacks the lymphoid marker CD7. Interestingly, the ILC3 (cluster 5) and ILC3-like (cluster 1) populations resided mainly in intestinal biopsies of patient with Crohn's disease (Fig. 3f) and may be related. Cluster 4 was mainly present in peripheral blood of patients with RCDII, suggesting a possible association with this pre-malignant disease state. Importantly, three clusters (4, 17, and

19) (Fig. 3f) were essentially missed in our previous study[14] due to the downsampling. Finally, all identified cell clusters consist to a variable extent of cells that were downsampled in the original analysis (Fig. 3g). Thus, the analysis of the full data set provides increased detail and confidence in establishing the phenotypes of these low abundance innate cell subsets.

**HSNE identifies rare CD4[+] T-cell subsets in blood**. Next, we selected the CD4[+] T-cell lineage (Fig. 3a) and show the distribution of the landmarks at the third level, revealing several clusters within the CD4[+] T-cell compartment (Fig. 5a), including a small CD28[−]CD4[+] T-cell memory population (25,398 cells; 0.5% of total), most likely representing terminally differentiated cells[26]. Subsequent analysis at the single-cell level (Fig. 5b) identified a CD56[+] population within the CD28[−]CD4[+] T cells that is enriched in blood of patients with Crohn's disease (Fig. 5b, bottom panels, dashed black circle), as well as a CD56[−] population of CD28[−]CD4[+] T cells (Fig. 5b, bottom panels, dashed yellow circle) present in blood samples of both patients and controls. Importantly, this latter cell population was not identified in our previous publication due to the non-uniform downsampling of cells (Fig. 5b).

Together, these findings emphasize that Cytosplore[+HSNE] is highly efficient in unbiased analysis of both abundant and rare cell populations in health and disease by permitting full single-cell

resolution. It enables the simultaneous identification and visualization of known cell subsets and provides evidence for additional heterogeneity in the immune system, as it reveals the presence of cell clusters that were missed in a previous analysis due to downsampling of the input data. These currently unspecified cell clusters might represent intermediate stages of differentiation or novel rare cell types with presently unknown function.

**HSNE is robust and outperforms current single-cell methods.** While the exploration of the hierarchy requires analysis at multiple levels, the workflow is robust and reproducible as shown in Supplementary Fig. 5. In this exemplary analysis, we obtained the same Cytosplore[+HSNE] clusters at the single-cell level upon reconstructing the hierarchy and embeddings in a matter of minutes (Methods section). In addition, we tested the Cytosplore[+HSNE] applicability to three different public mass cytometry data sets. First, we analyzed a well-characterized bone marrow data set[27] containing 81,747 cells as a benchmark case (Supplementary Fig. 6) and demonstrated that the landmarks in the overview level (2632; 3.2% of total) that were selected by the HSNE algorithm were distributed across almost all of the manually gated cell types (Supplementary Fig. 6a), indicating that the global data heterogeneity was accurately preserved. Also, GMS clustering resulted in HSNE clusters that were phenotypically similar to the manually gated cell types and displayed additional diversity within those subsets (Supplementary Fig. 6b). However, as the power of Cytosplore[+HSNE] lies in its scalability to data sets exceeding millions of cells, we also tested the versatility of Cytosplore[+HSNE] by comparing it to other state-of-the-art scalable single-cell analysis methods and accompanying large data sets (Supplementary Note 3. Cytosplore[+HSNE] offers advantages over current scalable single-cell analysis methods, Supplementary Figs. 7 and 8). Here Cytosplore[+HSNE] computed the analyses of the VorteX data set[5] containing 0.8 million cells in 4 min compared to 22 h, using the publicly available VorteX implementation on the same computer. Similarly, analysis of the Phenograph data set[4] containing 15 million cells was computed in 3.5 h compared to 40 h, using the publicly available Phenograph implementation on the same computer. Both analyses show that Cytosplore[+HSNE] reproduces the main findings as presented in the original publications. More importantly, Cytosplore[+HSNE] provides the distinct advantage of visualizing all cells and intracluster heterogeneity at subsequent levels of detail up to the single-cell level, even for the 15 million of cell data set, without a need for downsampling. Also, VorteX failed computing the 5.2 million cell gastrointestinal data set within 3 days of clustering (regardless of using Euclidian or Angular distance), where Cytosplore[+HSNE] accomplished this within 29 min. Moreover, while Phenograph did identify rare clusters that largely consisted of CD56[+] cells within the CD28[−]CD4[+] memory T cells (Fig. 5b), these clusters did not accurately correspond to the total number of CD56[+] cells, obscuring the association with Crohn's disease, further highlighting the advantages of Cytosplore[+HSNE] over these other computational tools.

Finally, we investigated whether a density-based downsampling as implemented for instance by SPADE[2], could provide better results compared to random downsampling. However, solely applying density-based downsampling does not allow for quantitative analysis of the resulting sample, as different types of cells will be reduced by different amounts. To mitigate this problem, SPADE implements an elaborate pipeline of downsampling, clustering and subsequent upsampling to enable for such a comparison, while this is an inherent part of HSNE. Therefore, we made a direct comparison between density-based downsampling used in the SPADE pipeline[2] and HSNE of the same 5.2 million cells gastrointestinal data set. On the basis of the expression of major lineage markers (Fig. 3a), HSNE created six large clusters (Fig. 3b) in the two-dimensional space at the overview level where similar landmark cells group closely, laying out all the cells of one cluster very close to any other cell of the same cluster, but distant from the cells of the other clusters. The SPADE analysis on the same data (Supplementary Fig. 9) created a dendrogram where cells of one cluster are close to cells of other clusters, while in high-dimensional space, they could be dissimilar and far apart. Importantly, we compared the ability of the SPADE analysis to preserve rare cellular subsets with HSNE. Despite density-based downsampling, several SPADE nodes that were created displayed a mixture of different phenotypes (underclustering) as revealed by the single-cell resolution of a linked t-SNE analysis that we show for the CD56[+]CD4[+] T-cell node as an example (Supplementary Fig. 9b, node #1), while other SPADE nodes contained cells with overlapping phenotypes (overclustering) such as several myeloid cell populations (Supplementary Fig. 9c, nodes #2–5). In addition, rare subsets such as the CD28[−] subpopulations of CD4[+] memory T cells (Supplementary Fig. 9d) or the ILC-like clusters (Supplementary Fig. 9e) that we could identify with HSNE (Figs. 3 and 5) were in the resulting SPADE tree indistinguishable from other CD4[+] T cells or innate lymphocytes, respectively (shown by the overlapping distributions of cells from different nodes); this indicates that SPADE is less suitable for rare cell analysis. A similar problem was reported by Amir et. al., where leukemic cells were not separated from healthy cells in the SPADE tree[9]. Thus, combining the single-cell resolution with the enhanced scalability may be critical for the success of HSNE in preserving rare cells.

## Discussion

Mass cytometry data sets generally consist of millions of cells. Current tools can either extract global information with no single-cell resolution or provide single-cell resolution but at the expense of the number of cells that can be analyzed. Consequently, when single-cell resolution is of interest, most current tools require downsampling of the data sets. However, reducing the number of included cells in the analysis pipeline may hamper the identification of rare subsets.

To overcome this problem, we introduce Cytosplore[+HSNE]. On the basis of a novel hierarchical embedding of the data (HSNE), Cytosplore[+HSNE] enables the analysis of tens of millions of cells using the whole data in a fraction of the time required by currently available tools. The power of the hierarchical embedding strategy is that Cytosplore[+HSNE] provides visualizations of the data at different levels of resolution, while preserving the non-linear phenotypic similarities of the single cells at each level. Cytosplore[+HSNE] enables the user to interactively select the groups of data points at each resolution level, either hand-picked or guided by density-based clustering, to further zoom-in on the underlying data points in the hierarchy up to the single-cell resolution. Using a data set of 5.2 million cells, we demonstrate that Cytosplore[+HSNE] allows a rapid analysis of the composition of the cells in the data set that, at all levels of the hierarchy, the representation of these cells preserve phenotypic relationships, and that one can zoom-in on rare cell populations that were missed with other analysis tools. The identification of such rare immune subsets offers opportunities to determine cellular parameters that correlate with disease.

There is an ongoing scientific debate on the validity of clustering in t-SNE maps versus direct clustering on the high-dimensional space. However, it has been shown that stochastic neighbor embedding (SNE) preserves and separates clusters in the high dimensional space[28]. While clustering the data points on highly non-linear manifolds is possible with complex models, we argue that the presented approach simplifies clustering

considerably. We show that HSNE efficiently unfolds the non-linearity in the high-dimensional data, as other SNE approaches do and therefore simpler clustering methods based on locality in the map suffice to partition the data faithfully (e.g., the density-based GMS clustering, implemented in Cytosplore[+HSNE]). Especially when combined with an interactive quality control mechanism to visually inspect residual variance within each cluster, the kernel size can be selected such that within-cluster variance is minimized, and thereby supports the validity of the cluster with respect to potential underclustering. This is indeed confirmed by comparisons to other scalable tools (i.e., Phenograph and VorteX), showing that Cytosplore[+HSNE] provides a superior discriminatory ability to identify and visualize rare phenotypically distinct cell clusters in large data sets in a very short time span. However, depending on user preference, Cytosplore[+HSNE] can be used in conjunction with such direct clustering approaches. This allows the user to identify additional heterogeneity that is potentially missed by direct clustering, and provides the tools for an informed merging and splitting of clusters as the user deems appropriate. The recent application of mass cytometry and other high-dimensional single-cell analysis techniques has greatly increased the number of phenotypically distinct cell clusters within the immune system. This raises obvious questions about the true distinctiveness and function of such cell clusters in health and disease, an issue that is beyond the scope of the present study but needs to be addressed in future studies.

In conclusion, Cytosplore[+HSNE] allows an interactive and fast analysis of large high-dimensional mass cytometry data sets from a global overview to the single-cell level and is coupled to patient-specific features. This may provide crucial information for the identification of disease-associated changes in the adaptive and innate immune system which may aid in the development of disease- and patient-specific treatment protocols. Finally, Cytosplore[+HSNE] applicability goes beyond analyzing mass cytometry data sets as it is able to analyze any high-dimensional single-cell data set.

## Methods

**HSNE algorithm**. HSNE builds a hierarchy of local and non-linear similarities of high-dimensional data points[13], where landmarks on a coarser level of the hierarchy represent a set of similar points or landmarks of the preceding more detailed level. To represent the non-linear structures of the data, the similarity of these landmarks is not described by Euclidean distance, but by the concept of AoI on landmarks of the preceding level. The similarities described in every level of the hierarchy are then used as input for an adapted version of the similarity-based embedding technique BH-SNE[15] for visualization.

The algorithm works as follows: First, a weighted k-nearest neighbor (kNN) graph is computed from the raw input data. For optimal performance and scalability, the neighborhoods are approximated as described in ref. [16]. The weight of the link between the two data points in the kNN graph describes the similarity of the connected data points.

In the subsequent steps, the hierarchy is built based on the similarities of the data level. To this extent, a number of random walks of predefined length is carried out starting from every node in the kNN graph, using the similarities as probability for the next jump; similar nodes to the current node are more likely to be the target of the next jump. Nodes in the graph that are reached more often are considered more important and selected as landmarks for the next coarser level. The number of landmarks is selected in a data-driven manner, based on this importance. The AoI of a landmark is defined by a second set of random walks started from all nodes (data points or landmarks on the preceding level). Here, the length is not predefined. Rather, once a landmark is reached, the random walk terminates. The influence on the node is then defined for every reached landmark as the fraction of walks that terminated in that landmark. Inversely, the AoI for each landmark is defined as the set of all nodes that reached this landmark at least once in this second set of random walks. Consequently, since multiple random walks initiated at the same node can end in different nodes, the AoIs of different landmarks can overlap.

We use this overlap to define a new neighborhood graph at the levels above the data level. Here, two nodes in the graph corresponding to landmarks at this level are connected if they have overlapping AoIs, where the link between the nodes is weighted by the number of data points in the overlapping area. This process is carried out iteratively, until a predefined number of hierarchical levels has been constructed. For the full technical details, we refer to our previous work[13].

**HSNE implementation in Cytosplore[+HSNE]**. We implemented our integrated analysis tool Cytosplore[+HSNE] using a combination of C++, javascript and OpenGL. All computationally demanding parts are implemented in C++ and make use of parallelization, where possible. The density estimation and GMS clustering make use of the graphics processing unit (GPU), as described in our original publication on Cytosplore[29], if possible, allowing clustering of millions of points in less than a second. We implemented the visualizations of the embedding in OpenGL on the GPU, for optimal performance, and less computational demanding visualizations, such as the heatmap, in javascript. We implemented the HSNE algorithm in C++, as presented in ref. [13]. Since we use the sparse data structures, memory consumption strongly depends on the data complexity. Maximum memory consumption during the construction of a four level hierarchy plus overview embedding of the 841,644 cell VorteX data set was 1,684 MB, construction of a five-level hierarchy of our human inflammatory intestinal diseases data set, consisting of 5,220,347 cells required a maximum of 9,357 MB of main memory, and finally, the 15,299,616 cell Phenograph data set required a maximum of 24.3 GB of memory during the computation of a five-level hierarchy plus the overview embedding. Computation times for the described hierarchies plus the first level embedding after 1,000 iterations were 4 min, 29 min, and, 3 h and 37 min, respectively, on a HP Z440 workstation with a single intel Xeon E5-1620 v3 CPU (4 cores) clocked at 3.5 Ghz, 64 GB of main memory and an nVidia Geforce GTX 980 GPU with 4 GB of memory, running Windows 7.

**Human gastrointestinal disorders mass cytometry data set**. Detailed description of the mass cytometry data set on human gastrointestinal disorders can be found in our previous work[14]. In brief, samples ($N = 102$) were collected from patients who were undergoing routine diagnostic endoscopies. The cells from the epithelium and lamina propria were isolated from two or three intestinal biopsies by treatment with EDTA followed by a collagenase mix under rotation at 37 °C. We analyzed single-cell suspensions from biological samples including duodenum biopsies ($N = 36$), rectum biopsies ($N = 13$), perianal fistulas ($N = 6$), and PBMC from control individuals ($N = 15$) and from patients with inflammatory intestinal diseases (celiac disease (CeD), $N = 13$; RCD type II (RCDII), $N = 5$; enteropathy-associated T-cell lymphoma type II (EATLII), $N = 1$ and Crohn's disease (Crohn), $N = 10$). A CyTOF panel of 32 metal isotope-tagged monoclonal antibodies was designed to obtain a global overview of the heterogeneity of the innate and adaptive immune system. Primary antibody metal-conjugates were either purchased or conjugated in-house. Procedures for mass cytometry antibody staining and data acquisition were carried out as previously described[27]. CyTOF data were acquired and analyzed on-the-fly, using dual-count mode and noise-reduction on. All other settings were either default settings or optimized with a tuning solution. After data acquisition, the mass bead signal was used to normalize the short-term signal fluctuations with the reference EQ passport P13H2302 during the course of each experiment and the bead events were removed[30].

**Processing of mass cytometry data**. We transformed data from the human inflammatory intestinal diseases data set using hyperbolic arcsin with a cofactor of 5 directly within Cytosplore[+HSNE]. We discriminated live, single CD45[+] immune cells with DNA stains and event length for the human inflammatory intestinal diseases study. We analyzed other data (Phenograph and VorteX data sets) as was available, except the transformation using hyperbolic arcsin with a cofactor of 5.

**Cytosplore[+HSNE] analysis**. Cytosplore[+HSNE] facilitates the complete exploration pipeline in an integrated manner (see Supplementary Movie 1). All presented tools are available for every step of the exploration and every level of the hierarchy. Data analysis in Cytosplore[+HSNE] included the following steps: We applied the arcsin transform with a cofactor of five upon loading the data sets. After that, we started a new HSNE analysis and defined the markers that should be used for the similarity computation. We used markers CD3, CD4, CD7, CD8a, CD8b, CD11b, CD11c, CD14, CD19, CD25, CD27, CD28, CD34, CD38, CD45, CD45RA, CD56, CD103, CD122, CD123, CD127 CD161, CCR6, CCR7, c-KIT, CRTH2, IL-15Ra, IL-21R, NKp46, PD-1, TCRab, and TCRgd for the human inflammatory intestinal diseases data set, all available markers for the bone marrow benchmark dataset, surface markers CD3, CD7, CD11b, CD15, CD19, CD33, CD34, CD38, CD41, CD44, CD45, CD47, CD64, CD117, CD123 and HLA-DR for the Phenograph dataset, and markers CD3, CD4, CD5, CD8, CD11b, CD11c, CD16/32, CD19, CD23, CD25, CD27, CD34, CD43, CD44, CD45.2, CD49b, CD64, CD103, CD115, CD138, CD150, 120g8, B220, CCR7, c-KIT, F4/80, FceR1a, Foxp3, IgD, IgM, Ly6C, Ly6G, MHCII, NKp46, Sca1, SiglecF, TCRb, TCRgd and Ter119 to construct the hierarchy for the VorteX data set. We used the standard parameters for the hierarchy construction; number of random walks for landmark selection: $N = 100$, random walk length: $L = 15$, number of random walks for influence computation: $N = 15$. For any clustering that occurred the GMS grid size was set to $S = 256$ ref. [2]. The reduction factor from one level in the hierarchy to the next coarser level is completely data-driven. In our experiments with mass cytometry data, the number of landmarks was consistently reduced by roughly one order of magnitude from one level to the next. Embeddings consisting of only a few hundred points usually provide little insight. Therefore, we defined the number of levels such that the overview level could be expected to consist of in the order of 1,000 landmarks

meaning $N = 5$ for the human inflammatory intestinal diseases data set and Phenograph data set, $N = 3$ for the bone marrow benchmark data set, and $N = 4$ for the VorteX data set. Building the hierarchy automatically creates a visualization of the overview level using BH-SNE. Cytosplore$^{+HSNE}$ enables color coding of the landmarks using expression (e.g., Fig. 3a) of any provided markers or by sample. For example, we created the clinical feature (e.g., Fig. 3c, bottom-left panel) and blood/ intestine (e.g., Fig. 3c, bottom-right panel) color schemes based on samples for the human inflammatory intestinal diseases data set within Cytosplore$^{+HSNE}$, and for the Phenograph data set, we created a color scheme that represented the sample coloring as provided in ref. [4] (Supplementary Fig. 7). For zooming into the data, we generally selected cells based on visible clusters, either using manual selection or by selecting clusters derived by using the GMS clustering. For the VorteX data set, we clustered the third level embedding (Supplementary Fig. 8). We specified a kernel size of 0.18 of the embedding size, to match the 48 clusters created by the X-shift clustering described in ref. [5], resulting in 50 clusters.

For subset classification, we first cluster the embedding at a given level using the GMS clustering. Next, we inspect the clustering by using the integrated descriptive marker statistics and heatmap visualization. If there is still meaningful variation of the marker expression within clusters, we zoom further into these clusters. If clusters are phenotypically homogeneous, the corresponding cell types are defined by inspecting the full marker expression profile in the heatmap and then the cluster is exported from any level in the hierarchy.

**Data availability**. The gastrointestinal mass cytometry data set that supports the findings of this study is publicly available on Cytobank, experiment no 60564. https://community.cytobank.org/cytobank/experiments/60564. The source code of the HSNE library, written in C++, is available at https://github.com/Nicola17/ High-Dimensional-Inspector. Furthermore, we provide a Cytosplore$^{+HSNE}$ installer for Windows, allowing exploration of several million cells, for academic use at https://www.cytosplore.org.

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

## Acknowledgements

The research leading to these results has received funding from Leiden University Medical Center, the Netherlands Organization for Scientific Research (ZonMW grant 91112008) and the Technology Foundation STW, the Netherlands (VAnPIRe; grant 12720, and Genes in Space; grant 12721). We thank Drs M.W. Schilham, M. Yazdanbakhsh, J. Goeman, K. Schepers, J. van Bergen and S.E. de Jong for critical review of the manuscript and B. van Lew for narrating the Supplementary Movie 1.

## Author contributions

V.v.U., T.H., N.P., F.K., A.V. and B.P.F.L.: Conceived the study. T.H., N.P., A.V. and B.P.F.L.: Developed the HSNE method and implementation in Cytosplore$^{+HSNE}$. V.v.U. and F.K.: Performed the biological analysis and interpretation. T.H.: Performed the t-SNE scalability analysis and comparison. V.v.U.: Performed the hierarchy robustness analysis. V.v.U. and T.H.: Performed the comparison with other methods. N.L., M.J.T.R. and E.E.: Provided conceptual input. V.v.U., T.H., N.P., F.K., A.V. and B.L.: Wrote the manuscript. All authors discussed the results and commented on the manuscript.

## Additional information

**Competing interests:** The authors declare no competing financial interests.

**Reprints and permission** information is available online at http://npg.nature.com/ reprintsandpermissions/

