## [Peer Review File · Nature Communications]

Reviewers' comments:

Reviewer #1 (Remarks to the Author):

van Unen et al. present an implementation of Hierarchical Stochastic Neighbor Embedding, a non-linear dimensionality reduction technique that presents the advantage of aggregating chunks of the data at various levels of granularity. This aggregation allows for a rapid computation of BH-SNE that can be used to interactively display and explore the data. The authors provide convincing data about the validity of their approach to represent the data (in the sense that it provides results of similar accuracy as compared to BH-SNE while requiring a much-reduced computational time, removing the need for downsampling). The authors went through great efforts to implement their algorithm efficiently and to implement it in a user-friendly software platform. The results are nicely described. This method appears to be very useful and could possibly make a large impact on the field of single cell analysis. The subject of the current paper, HSNE, is however very much linked to their software platform, Cytosplore, which appears to make HSNE very useful. However, it would also be good for the community to make sure that users can easily port HSNE to other platforms if they want to, and thus I believe the C code should be made public so that it can be interfaced with R / Python.

Comments:

As mentioned above, it seems important for the authors to provide code for this algorithm so that it can be used outside of the Cytosplore platform. Without this capacity, it will be very difficult to implement this algorithm in other workflows or applications.

Line 44: "However, the linear nature of PCA renders it unsuitable to dissect the non-linear relationships in mass cytometry data, while the non-linear methods (t-SNE8 and Diffusion maps10) do retain local data structure, but are limited by the number of cells that can be analyzed."

I think the problem is rather that linear dimensionality reduction will inevitably run into a crowding problem when represented in two dimensions, so non-linear techniques are inherently necessary for exploratory analyses of CYToF data.

Line 95: "We recommend to use $\log_{10}(N/100)$ levels" : any comment on how this was arrived?

Line 167: "We selected 145 clusters using GMS clustering at the third level and observed that the identified clusters contained variable numbers of downsampled and non-classified cells (Fig. 2d)." I expect these proportions to be most meaningful for large clusters. Is there a correlation between the size of the 145 clusters and their % mapped?

Line 267: "Finally, while Phenograph did identify the rare CD56+ population of CD28-CD4+ T cells in the peripheral blood of individual samples (Fig. 4b), it did not reveal the association with Crohn's disease, further highlighting the advantages of Cytosplore+HSNE over these other computational tools." What exactly do the authors imply here? I find it hard to believe that the "association" is hidden if the same cluster is identified. Obviously Phenograph, being a clustering method, won't directly reveal the associations with phenotypes as its output but highly facilitate their discovery.

Line 374: "Code availability." I'd argue that this header should be renamed "Software availability" as the source code is not open. See above about the importance of making the code accessible.

I do not understand what data points are actually selected when one selects a set of landmarks in Cytosplore. Is it e.g. all the data points whose biggest influence is in the set of landmarks? The methods describe that data points are used to create a kNN graph, then random walk to identify landmark points, then data points are used to compute AoI of each landmark. But I don't understand how to go from there to the next iteration. If the above is correct, can a subset of the

cells be mis-classified in the high-level hierarchy, and thus "contaminate" lower levels?

Can the authors explain what actually happen when a user "drills in" ?

On Sup Fig 6b, I believe the labels for Eosinophils and Basophils are swapped.

The software seems really useful and user-friendly. Are there some obvious misuses the authors want to warn potential users against? It is quite common to see mis- or over-interpreted t-SNE even in peer-reviewed publications. Are there additional potential pitfalls that users should be warned about that are specific to HSNE?

Can Cytosplore users export the output of hSNE? I.e. the set of landmark points and their hierarchical levels, their AoI across data points?

Reviewer #2 (Remarks to the Author):

The manuscript presents a new method called HSNE for analyzing large CyTOF datasets. It performs a hierarchical tSNE by incorporating cluster analysis. HSNE improves time efficiency and overcomes the limitations of downsampling. It also provides an interactive and hierarchical visualization of cellular complexity. The manuscript also demonstrates the utilities of HSNE on several datasets.

Some comments:

1. A comparison between HSNE and a more intelligent downsampling will be helpful. Random subsampling may discard rare cell populations. Density-based downsampling such as the one used in SPADE pipeline may be able to preserve rare cells.
2. In most of the figures, tSNE plot at data level appears to be a big round shape. It could be due to the fact that the cells are quite homogeneous. From our experience, when a dataset contains a high number of very similar cells, tSNE generates a big round shape. Would HSNE be able to tackle this issue?
3. Due to the tSNE performance mentioned above, the cluster partitions (for example Figure 3E) are not very convincing. Different clusters are very close to each other.
4. At the overview level, I am wondering whether there is a way to also illustrate how heterogeneous the cells in(or associated to) individual landmarks are.

Response to Reviewers

Reviewer #1 (Remarks to the Author):

van Unen et al. present an implementation of Hierarchical Stochastic Neighbor Embedding, a
non-linear dimensionality reduction technique that presents the advantage of aggregating
chunks of the data at various levels of granularity. This aggregation allows for a rapid
computation of BH-SNE that can be used to interactively display and explore the data. The
authors provide convincing data about the validity of their approach to represent the data (in the
sense that it provides results of similar accuracy as compared to BH-SNE while requiring a
much-reduced computational time, removing the need for downsampling). The authors went
through great efforts to implement their algorithm efficiently and to implement it in a user-friendly
software platform. The results are nicely described. This method appears to be very useful and
could possibly make a large impact on the field of single cell analysis. The subject of the current
paper, HSNE, is however very much linked to their software
platform, Cytosplore, which appears to make HSNE very useful. However, it would also be good
for the community to make sure that users can easily port HSNE to other platforms if they want
to, and thus I believe the C code should be made public so that it can be interfaced with R /
Python.

Comments:

As mentioned above, it seems important for the authors to provide code for this algorithm so
that it can be used outside of the Cytosplore platform. Without this capacity, it will be very
difficult to implement this algorithm in other workflows or applications.

We agree and will make an open source library written in C++ for HSNE analysis publicly
available at <https://github.com/Nicola17/High-Dimensional-Inspector> upon publication of the
manuscript, in addition to making Cytosplore^{+HSNE} available in an executable application form.

Line 44: "However, the linear nature of PCA renders it unsuitable to dissect the non-linear
relationships in mass cytometry data, while the non-linear methods (t-SNE8 and Diffusion
maps10) do retain local data structure, but are limited by the number of cells that can be
analyzed."

I think the problem is rather that linear dimensionality reduction will inevitably run into a
crowding problem when represented in two dimensions, so non-linear techniques are inherently
necessary for exploratory analyzes of CYToF data.

Both linear and non-linear techniques suffer from crowding problems with increasing data sizes.
The main issue here is that mass cytometry data cannot be dissected clearly with linear
techniques, even for small datasets. Consider a dataset consisting of cells from multiple
immune lineages, with strongly different expression values over many markers. A linear
technique will place the cells from the different lineages far apart, and therefore will not provide

enough space for cells with smaller differences within the lineages. On the other hand, a non-
linear technique such as t-SNE or HSNE will highlight the small differences, while largely
different cells will still be located in separate groups, however, not placed as far apart as their
linear differences might suggest.

Line 95: "We recommend to use $\log_{10}(N/100)$ levels" : any comment on how this was arrived?

We added additional explanation in the manuscript. In short, we found that the data is reduced
by roughly one order of magnitude per HSNE level with mass cytometry data (the reduction is
data-driven, so we cannot make a hard definition). Since the Barnes-Hut algorithm works
optimally with a few thousand data points, $\log_{10}(N/100)$ levels will yield an optimal number of
landmarks at the overview level. It should also be noted that constructing an HSNE hierarchy
using more levels is possible. If the resulting embedding at the highest overview level is
uninformative, the user can simply request the next, more detailed level of the hierarchy without
partitioning by selecting all the embedded landmarks to zoom in.

Line 167: "We selected 145 clusters using GMS clustering at the third level and observed that
the identified clusters contained variable numbers of downsampled and non-classified cells (Fig.
2d)." I expect these proportions to be most meaningful for large clusters. Is there a correlation
between the size of the 145 clusters and their % mapped?

Both large and small size HSNE clusters contained variable numbers of discarded cells, and
there does not appear to be a correlation (See Response Fig. 1 below). In this particular study¹,
it was decided to equal the number of cells from blood and intestinal samples for a balanced
comparison, which led to the exclusion of more cells from the blood samples. The following
analysis on the downsampled dataset had revealed that there are immune-system-wide
differences in subset composition between peripheral blood and intestinal samples dividing
them into two large sample clusters. Therefore, naturally the HSNE clusters containing
predominantly blood cells tended to contain more discarded cells than those containing
intestinal cells (See Response Fig. 1 below). Overall, the extent of discarded cell frequencies
within HSNE clusters was independent on their respective cell size, and in this particular study
was most abundant but not exclusive to those containing predominantly cells from peripheral
blood.

 **Response Figure 1. Cell size of HSNE clusters do not correlate with frequency of discarded cells.**
 (Top panel) Frequency of annotated cells for 145 clusters identified by HSNE (adapted from Fig. 2d).
 Color represents the subsets (N=142), as identified in a previous study¹. Black represents the cells
 discarded by stochastic downsampling and grey the cells discarded by ACCENSE clustering. (Bottom
 panels) Corresponding cell size of the HSNE clusters (% of total 5.2 million cells; black-to-yellow color)
 and predominant tissue-origin of cells within HSNE clusters (red and blue) are depicted in a heatmap
 below.

 Line 267: "Finally, while Phenograph did identify the rare CD56⁺ population of CD28⁻CD4⁺ T
 cells in the peripheral blood of individual samples (Fig. 4b), it did not reveal the association with
 Crohn's disease, further highlighting the advantages of Cytosplore+HSNE over these other
 computational tools." What exactly do the authors imply here? I find it hard to believe that the
 "association" is hidden if the same cluster is identified. Obviously Phenograph, being a
 clustering method, won't directly reveal the associations with phenotypes as its output but highly
 facilitate their discovery.

 We thank the reviewer for pointing this out. Actually the phrasing of the sentence was not
 entirely correct. A side by side comparison of the CD56 expression and the Phenograph
 clusters (Response Figure 2 below) indicates that Phenograph does identify two (overlapping)
 CD56⁺ clusters of cells (yellow and blue in right panel) but these clusters do not accurately
 correspond to all of the CD56⁺ cells. Here part of the CD56⁺ cells are missed (red circle in right
 panel) and those are mixed with CD56⁻ cells, making it more difficult to appreciate the
 association with Crohn's disease.

 We have now rephrased the sentence to: "Moreover, while Phenograph did identify rare clusters
 that largely consisted of CD56⁺ cells within the CD28⁻CD4⁺ memory T cells, these clusters did
 not accurately correspond to the total number of CD56⁺ cells, obscuring the association with
 Crohn's disease, further highlighting the advantages of Cytosplore^{+HSNE} over these other
 computational tools."

Response Figure 2. Phenograph analysis of the gastrointestinal dataset.

The 5.2 million cell gastrointestinal dataset was analyzed by Phenograph. HSNE data level of CD28⁻ memory CD4⁺ T cells after clustering with Phenograph. Colors indicate arcsin5-transformed CD56 marker expression (coloring range: 0-5) (left) and Phenograph cluster identity (right). Red circle indicates CD56⁺ cells that were not present within the CD56⁺ yellow and blue Phenograph clusters

Line 374: "Code availability." I'd argue that this header should be renamed "Software availability" as the source code is not open. See above about the importance of making the code accessible.

Since we will also make the code available at the request of the reviewers we decided to change the name of the section to *Software and Code Availability*.

I do not understand what data points are actually selected when one selects a set of landmarks in Cytosplore. Is it e.g. all the data points whose biggest influence is in the set of landmarks?

We adjusted the phrasing in section *Interactive Exploration* to clarify that all landmarks that are present in the area of influence of the selected landmarks at the next HSNE level will be selected. While this is true for the implementation in Cytosplore the HSNE library that we will make available in addition allows different types of selections, including selecting the data points whose biggest influence is in the set of landmarks.

The methods describe that data points are used to create a kNN graph, then random walk to identify landmark points, then data points are used to compute Aol of each landmark. But I don't understand how to go from there to the next iteration.

We extended the description of the construction in the main text, as well as in the methods section, to clarify how the graph is constructed beyond the kNN graph at the data level.

For the levels beyond the data level we create the neighborhood graph based on the Aols. Two
nodes with overlapping Aols will be linked in the graph and the strength of the link is defined by
the amount of overlap. This effectively propagates the non-linear structures of the data to the
higher levels in the hierarchy.

If the above is correct, can a subset of the cells be mis-classified in the high-level hierarchy, and
thus "contaminate" lower levels?

In theory, the segmentation of the space can lead to such mis-classification. However, as we
show in Supplementary Figure 6, the aggregation strongly overlaps with traditional clustering
techniques. Furthermore, in the exploration phase the corresponding cells would show up as
separate groups at one of the more detailed levels. In combination with the linked heatmap they
could immediately be identified and properly classified.

Can the authors explain what actually happen when a user "drills in" ?

We used the term "drill in" in the supplementary movie synonymously to "Select and Zoom-In"
in the manuscript. We adjusted the voice over to make sure the same terminology is used in the
movie and the manuscript. As described above as a reply to "what data points are actually
selected when...", we extended the corresponding part in results section *Interactive Exploration*
to clarify what happens when a user selects and zooms in a set of landmarks in
Cytosplore^{+HSNE}.

On Sup Fig 6b, I believe the labels for Eosinophils and Basophils are
swapped.

The reviewer is correct. We apologize for this error and adapted the annotation of these cell
types appropriately in the revised Supplementary Information.

The software seems really useful and user-friendly. Are there some obvious misuses the
authors want to warn potential users against? It is quite common to see mis- or over-interpreted
t-SNE even in peer-reviewed publications. Are there additional potential pitfalls that users should
be warned about that are specific to HSNE?

The non-linear structure of the HSNE embeddings has similar pitfalls as any other non-linear
embedding technique such as t-SNE. Users need to be aware that, in contrast to linear
techniques such as PCA, large distances in the map do not possess a meaning. Other
properties of t-SNE, such as its tendency to split gradual transitions into separate clusters are
also apparent in HSNE and need to be taken into account when interpreting results.

Furthermore, while the ability of HSNE to maintain non-linearity and single-cell resolution
without the need for downsampling makes it outstanding for delineating the immune system
complexity and identifying rare cell types, the biological significance of the identified
heterogeneity is an issue that needs to be explored further. This relates to an ongoing debate
on the definition of a cell subset as the new analysis techniques including mass cytometry and

single-cell sequencing rapidly advance the number of “cell subsets” that can be identified.
HSNE analysis may accelerate the identification of distinct cell clusters. However, this does not
necessarily translate to functional differences as a related pair of clusters may also represent
two differentiation or intermediate stages of a single cellular subset.

Can Cytosplore users export the output of hSNE? I.e. the set of landmark points and their
hierarchical levels, their AoI across data points?

We envision Cytosplore as an integrated tool for the complete analysis pipeline and therefore do
provide several functions to save results. For example the user can save single-cell data of
clusters to fcs files, the cluster heatmaps can be exported as editable svg images and/or csv for
further numerical analysis, individual sample statistics for clusters can be exported as csv files,
images of the embeddings with all marker expression overlays can be saved in one step, and
images of the embeddings of sample (group) visualizations and density representations can
also be exported. The standalone HSNE library does support the output of the complete
hierarchy as imagined by the reviewer.

Reviewer #2 (Remarks to the Author):

The manuscript presents a new method called HSNE for analyzing large CyTOF datasets. It
performs a hierarchical tSNE by incorporating cluster analysis. HSNE improves time efficiency
and overcomes the limitations of downsampling. It also provides an interactive and hierarchical
visualization of cellular complexity. The manuscript also demonstrates the utilities of HSNE on
several datasets.

Some comments:

1. A comparison between HSNE and a more intelligent downsampling will be helpful. Random
subsampling may discard rare cell populations. Density-based downsampling such as the one
used in SPADE pipeline may be able to preserve rare cells.

We have followed the reviewer’s suggestion and made a direct comparison between density-
based downsampling used in the SPADE pipeline and HSNE of the same 5.2 million cells
gastrointestinal dataset. Based on the expression of major lineage markers (Fig. 3a) HSNE
created six large clusters (Fig. 3b) in the two-dimensional space at the overview level where
similar landmark cells group closely, laying out all the cells of one cluster very close to any other
cell of the same cluster, but distant from the cells of the other clusters. The SPADE analysis on
the same data (Response Fig. 3a, below) created a dendrogram where cells of one cluster are
close to cells of other clusters, while in high-dimensional space they could be dissimilar and far
apart.

Importantly, we compared the ability of the SPADE analysis to preserve rare cellular subsets
with HSNE. Despite density-based downsampling, several SPADE nodes that were created

displayed a mixture of different phenotypes (underclustering) as revealed by the single-cell
resolution of a linked t-SNE analysis that we show for the CD56⁺CD4⁺ T cell node as an
example (Response Fig. 3b, node #1), while other SPADE nodes contained cells with
overlapping phenotypes (overclustering) such as several myeloid cell populations (Response
Fig. 3c, nodes #2-5). In addition, rare subsets such as the CD28⁻ subpopulations of CD4⁺
memory T cells (Response Fig. 3d) or the ILC-like clusters (Response Fig. 3e) that we could
identify with HSNE (Figs. 3 and 4) were indistinguishable from other CD4⁺ T cells or innate
lymphocytes in the resulting SPADE tree (shown by the overlapping distributions of cells from
different nodes); indicating that SPADE is less suitable for rare cell analysis. A similar problem
was reported by Amir et. al² where leukemic cells were not separated from healthy cells in the
SPADE tree. Thus, combining the single-cell resolution with the enhanced scalability may be
critical for the success of HSNE in preserving rare cells.

We have incorporated the Response Figure 3 below as a new Supplemental Figure in the
results section of the revised manuscript.

Response Figure 3. SPADE analysis of the gastrointestinal dataset.

The 5.2 million cell gastrointestinal dataset was analyzed by SPADE using 10% target events for density-based downsampling and 142 target nodes as settings. (a) SPADE tree colored with arcsin5-transformed marker expression as indicated. Size of the nodes represents cell size. Five nodes are indicated for further analysis. (b) t-SNE embedding of node 1 (panel a) at single-cell resolution. Color of cells as in panel a. (c) A heatmap summary of median expression values of cell markers expressed by nodes 2-5

(panel a). Coloring as in panel a. (d) HSNE data level of CD28⁻ memory CD4⁺ T cells after clustering with
SPADE. Colors indicate marker expression (left) and SPADE node identity (right) (e) HSNE data level of
ILC and ILC-like clusters after clustering with SPADE. Coloring as in panel d.

2. In most of the figures, tSNE plot at data level appears to be a big round shape. It could be
due to the fact that the cells are quite homogeneous. From our experience, when a dataset
contains a high number of very similar cells, tSNE generates a big round shape. Would HSNE
be able to tackle this issue?

Indeed, the HSNE data-level plots seem rather homogeneous. In the examples in Figures 3 and
4, we zoomed into 6,000 and 2,500 cells (out of 5.2 million), respectively. Since these cells were
grouped by similarity in the first place it is clear that these embeddings contain a rather
homogeneous group of cells with gradual transitions instead of clear cut-offs. However,
overlaying the different markers still shows some clear separations within the embeddings.
Furthermore, while the point-based representation is great for visualizing the different markers,
it can be problematic to reveal the real structure of the embeddings, especially when differences
are characterized by transitions instead of clear cut-offs. Much lower density transitional areas
will often appear similar to high density cluster centers. Therefore we allow on-the-fly switching
to the density representation in Cytosplore, and also show these density plots in Figures 3 and
4. The density plots show the different clusters more clearly than the point-based
representation. In addition, the distinctness of each cluster can further be explored by the linked
heatmap view, as shown in Figure 3 where the clusters clearly display unique marker
expression profiles in combination with intra-cluster heterogeneity (Response Figure 4).

3. Due to the tSNE performance mentioned above, the cluster partitions (for example Figure 3E)
are not very convincing. Different clusters are very close to each other.

In general we agree with this assessment. Indeed these plots are dominated by smooth
transitions rather than clear cluster cut-offs, as described above. We do believe that this is one
of the strengths of an interactive visual system like Cytosplore over unsupervised clustering
methods, as the user can immediately inspect and adjust the clustering, whereas the result of
an unsupervised clustering algorithm might hide such problems and is often interpreted as the
ground truth. Being able to inspect the clusters and visualize the expression of all markers per
cell in the embedding plot allows the user to make an informed decision on whether the
clustering result is valid.

4. At the overview level, I am wondering whether there is a way to also illustrate how
heterogeneous the cells in(or associated to) individual landmarks are.

There are two features in Cytosplore that we have implemented to guide the user in inspecting
the heterogeneity of cell clusters at any level of the hierarchy:

1) Once a group of landmarks/cells has been selected (manually or by GMS clustering), a
list of descriptive statistics of marker expressions within a selection including median,
mean and standard deviation can be obtained by selecting *Extended Settings* (shown in
Supplementary Fig. 3)

2) In the Heatmap view, selecting *Show Variation* will reveal less paint in the box if a cluster
is heterogeneous for a marker (See Response Figure 4 below).

At the overview or intermediate HSNE levels, both these tools illustrate the heterogeneity of all
the cells associated to the individual landmarks. To make this point clearer in the manuscript,
we extended the text in the results section:

*“Moreover, interactively linked heatmap visualizations of clusters (Fig. 1c, right panels)*
*and descriptive statistics of markers within a selection can be used to guide the*
*exploration.”*

...with the following sentence:

*“For example, these tools allow to inspect the heterogeneity of cells within individual*
*clusters, including the cells associated to individual landmarks.”*

**Response Figure 4. Heatmap visualization of cluster heterogeneity.** A screenshot from the
Supplementary Movie 1 (timepoint: 07:06). Five clusters (left view) are visualized in the Heatmap view
(right). In addition to median marker display in color, the Heatmap allows for marker variance display
reflected by the amount of paint in the box. The more narrow the box the higher the variance of a marker
a cluster displays (for example, the leftmost cluster is heterogeneous for CD56 and CD38 expression).

**References**

- 1. van Unen, V. *et al.* Mass Cytometry of the Human Mucosal Immune System Identifies
Tissue- and Disease-Associated Immune Subsets. *Immunity* **44**, 1227–1239 (2016).
2. Amir, E.-A. D. *et al.* viSNE enables visualization of high dimensional single-cell data and
reveals phenotypic heterogeneity of leukemia. *Nat. Biotechnol.* **31**, 545–552 (2013).

REVIEWERS' COMMENTS:

Reviewer #1 (Remarks to the Author):

These responses and the intention to share source code are much appreciated. My concerns have been adequately addressed.

Reviewer #2 (Remarks to the Author):

The authors have addressed most of the comments in great details.

One minor comment:

1. The authors compared SPADE with HSNE and showed that SPADE clustering is less accurate compared to HSNE. However, SPADE uses its own different clustering algorithm, hence the authors are actually comparing SPADE clustering with HSNE clustering algorithms. Is it possible to first down-sample the data with SPADE's density-based downsampling approach and then apply tSNE and the same clustering algorithm as used in HSNE? This would make a fair comparison between HSNE and density-based downsampling.

Response to Reviewer

Reviewer 2 comment :

“The authors compared SPADE with HSNE and showed that SPADE clustering is less accurate compared to HSNE. However, SPADE uses its own different clustering algorithm, hence the authors are actually comparing SPADE clustering with HSNE clustering algorithms. Is it possible to first down-sample the data with SPADE's density-based downsampling approach and then apply tSNE and the same clustering algorithm as used in HSNE? This would make a fair comparison between HSNE and density-based downsampling.”

We have carefully analyzed the reviewer's request in that indeed, the SPADE uses a different clustering strategy that first uses density-based subsampling to reduce the data; However, during the density-based downsampling process, the relative quantities of different cell populations in the full data are distorted, making it difficult to quantify relative proportions of cell populations. To restore these quantitative cell population ratios, SPADE downsampling is therefore necessarily always augmented with a strategy to upsample the data again once the cluster phenotypes are established.

We would be willing to add a density-based downsampling, followed by embedding the samples with t-SNE alone, but it does not give a fair comparison with HSNE, because it does not preserve the quantitative traits of the cell types in the data, while HSNE does so.

Two other downsides of a one-shot (density-based) downsampling of the data are the following:

- 1) The multi-level structure of HSNE gradually reduces the number of landmarks while keeping the non-linear structures in the data, as explained in "Hierarchy Construction" and "Aggregation of Data" in the results section. A one-shot downsampling (independent of the approach) will always suffer from losing these connections, and
- 2) The density-based downsampling proposed in SPADE is of $O(N^2)$ computational complexity which makes it infeasible for very large data as presented. In the original SPADE publication the authors circumvent the problem by downsampling individual samples/fcs-files independently (We did the same for the SPADE comparison in the latest revision). Even with this 'trick' the procedure took several hours compared to minutes of computing the HSNE hierarchy.

Taken together, we are therefore under the impression that adding the suggested tSNE-based clustering after density-based subsampling to the manuscript may cause more confusion than help. Instead we propose to demarcate this issue in the section describing the comparison to the full SPADE analysis (Page 14) that we added in the previous revision with a short reasoning on why we compare to SPADE instead of other density-based downsampling approaches.

Specifically, to address this final reviewer comment, we added the following text on page 14 (highlighted below in yellow):

“Finally, we investigated whether a density-based downsampling as implemented for instance by SPADE, could provide better results compared to random downsampling. However, solely applying density-based downsampling does not allow for quantitative analysis of the resulting sample, as different types of cells will be reduced by different amounts. To mitigate this problem, SPADE implements an elaborate pipeline of downsampling, clustering and subsequent upsampling to restore quantitative traits of the data. Therefore, we made a direct comparison between density-based downsampling used in the SPADE pipeline² and HSNE of the same 5.2 million cells gastrointestinal dataset. Based on the expression of major lineage markers (Fig. 3a) HSNE created six large clusters (Fig. 3b) in the two-dimensional space at the overview level where similar landmark cells group closely, laying out all the cells of one cluster very

close to any other cell of the same cluster, but distant from the cells of the other clusters. The SPADE analysis on the same data (Supplementary Fig. 9) created a dendrogram where cells of one cluster are close to cells of other clusters, while in high-dimensional space they could be dissimilar and far apart. Importantly, we compared the ability of the SPADE analysis to preserve rare cellular subsets with HSNE.”

We hope that with this further clarification and modification, we have addressed the last concern of the reviewer and that the manuscript can now be accepted for publication.